# Relationship between Coma Recovery Scale-Revised and the Thalamocortical Tract of Ascending Reticular Activating System in Hypoxic–Ischemic Brain Injury: A Pilot Study

**DOI:** 10.3390/healthcare11081148

**Published:** 2023-04-17

**Authors:** Sungho Jang, Eunbi Choi

**Affiliations:** Department of Physical Medicine and Rehabilitation, College of Medicine, Yeungnam University, Daegu 42415, Republic of Korea

**Keywords:** hypoxic–ischemic brain injury, Coma Recovery Scale-Revised, thalamocortical tract, diffusion tensor tractography, diffusion tensor imaging

## Abstract

Background: This pilot study examined the relationship between the Coma Recovery Scale-Revised (CRS-R) and the five subparts of the thalamocortical tract in chronic patients with hypoxic–ischemic brain injury by diffusion tensor tractography (DTT). Methods: Seventeen consecutive chronic patients with hypoxic–ischemic brain injury were recruited. The consciousness state was evaluated using CRS-R. The five subparts of the thalamocortical tract (the prefrontal cortex, the premotor cortex, the primary motor cortex, the primary somatosensory cortex, and the posterior parietal cortex) were reconstructed using DTT. Fractional anisotropy and the tract volume of each subpart of the thalamocortical tract were estimated. Results: The CRS-R score showed a moderate positive correlation with the tract volume of the prefrontal cortex part of the thalamocortical tract (*p* < 0.05). In addition, the tract volume of the prefrontal cortex component of the thalamocortical tract could explain the variability in the CRS-R score (*p* < 0.05). Conclusion: The prefrontal cortex part was closely related to the CRS-R score in chronic patients with hypoxic–ischemic brain injury. In addition, the change in the remaining number of neural fibers of the prefrontal cortex part appeared to be related to the change in conscious state.

## 1. Introduction

Disorders of consciousness (DOC) are clinical conditions caused by various brain pathologies, including hypoxic–ischemic brain injury [1,2]. DOC are the foremost sequelae of hypoxic–ischemic brain injury; more than 30% of patients who survived hypoxic–ischemic brain injury were reported to exhibit DOC [1,2]. For the control of consciousness, various neural structures of the brain play an important role, especially the prefrontal cortex, basal forebrain, anterior cingulate cortex, posterior occipital cortex, and claustrum [3,4]. However, it is not clear which of the brain regions are responsible for the recovery of impaired consciousness after hypoxic–ischemic brain injury. Detailed information on the neural structures related to consciousness in DOC patients with hypoxic–ischemic brain injury can be useful when establishing effective neuro-rehabilitation or neurointervention therapeutic strategies [5,6,7,8]. In particular, recently developed non-invasive brain stimulation therapies, such as repetitive transcranial magnetic stimulation or transcranial direct current stimulation, are being used to apply targeted stimulation to specific neural structures in order to facilitate recovery [5,6,7,8].

To evaluate the conscious state of patients with DOC, the Coma Recovery Scale-Revised (CRS-R), which is strongly recommended by the American Congress of Rehabilitation Medicine task force, is the most widely used behavioral assessment tool in clinical care and research in patients with subacute-to-chronic DOC [5,9]. In addition, a close relationship between the CRS-R and the entire thalamocortical tract between the thalamus and the cerebral cortex has also been reported [10,11,12,13]. However, little is known about the relationship between the CRS-R and the specific brain area within the thalamocortical tract responsible for consciousness.

The neural structures for controlling consciousness consist of various neural networks or connections, including the default mode network, ascending reticular activating system, thalamocortical connectivity, temporoparietal connectivity, frontoparietal connectivity, and cortical effective connectivity [14,15,16,17]. The thalamocortical tract of the ascending reticular activating system, which connects the thalamic intralaminar nuclei to the frontoparietal cortex, is considered an important neural pathway for controlling consciousness, particularly awareness rather than alertness [13,18]. To better understand the organization of the thalamocortical tract within the ascending reticular activating system, the thalamocortical tract can be divided into five distinct subparts that correspond to specific regions of the cortex. The subparts include the parts between the thalamic intralaminar nuclei and the prefrontal cortex, the premotor cortex, the primary motor cortex, the primary somatosensory cortex, and the posterior parietal cortex [19]. Many studies have reported that various brain areas are related to the CRS-R in patients with DOC using positron emission tomography, functional magnetic resonance imaging, and diffusion tensor imaging [20,21,22,23,24,25,26,27]. However, a precise estimation of the entire thalamocortical tract or subparts of the thalamocortical tract has been limited by the difficulty in discriminating it from adjacent neural structures on the above brain imaging methods.

In contrast, the recently developed diffusion tensor tractography (DTT), derived from diffusion tensor imaging, can visualize and estimate the thalamocortical tract of the ascending reticular activating system in three dimensions [19,28,29]. The state of the thalamocortical tract can be determined by evaluating the DTT parameters, such as the fractional anisotropy and the tract volume [28,29]. A few studies demonstrated the close relationship between the CRS-R and the entire thalamocortical tract between the thalamus and cerebral cortex in DOC patients with a range of brain pathologies using DTT [10,11,12,13]. However, no studies on the relationship between the CRS-R and the subparts of the thalamocortical tract have been reported. Therefore, the CRS-R would be related to a specific subpart of the thalamocortical tract in patients with hypoxic–ischemic brain injury. We hypothesized that the DTT parameters of a specific subpart of the thalamocortical tract could be related to the CRS-R score in patients with hypoxic–ischemic brain injury. This pilot study examined the relationship between the CRS-R and the five subparts of thalamocortical tract in chronic patients with hypoxic–ischemic brain injury using DTT.

## 2. Materials and Methods

### 2.1. Subjects

Seventeen consecutive patients (nine males and eight females; mean age 47.18 ± 19.01 years, range of 18–76 years) with hypoxic–ischemic brain injury were recruited. The patients were admitted to the rehabilitation department of a university hospital during the period of 2018 to 2021. The participants have an obvious hypoxic–ischemic brain injury history, such as cardiac arrest, strangulation, and carbon monoxide intoxication. The age of onset of hypoxic–ischemic brain injury ranged from 18 to 79 years, and all patients exhibited impaired consciousness from the onset of hypoxic–ischemic brain injury. For the inclusion criteria, the patients with no prior history of head trauma or neurological or psychiatric disease were selected. Diffusion tensor imaging was obtained during the chronic stage of hypoxic–ischemic brain injury, which was more than three months after onset. The demographic data of the patients, including age, sex, duration to diffusion tensor imaging, and consciousness state, are summarized in Table 1. This study was conducted retrospectively. The study protocol was approved by the institutional review board of a university hospital.

### 2.2. Clinical Evaluation

The consciousness state was evaluated using the CRS-R at the time of diffusion tensor imaging scanning. The evaluator was blinded to the results of the DTT analysis. The CRS-R is a widely used and standardized neurobehavioral assessment measure, used for patients with DOC [5]. The CRS-R consists of six subscales, which are auditory, visual, motor, oromotor/verbal, communication, and arousal subscales. The assessment score is obtained based on the patient’s responsiveness to stimuli and the presence of certain behaviors [5]. The auditory subscale evaluates the patient’s response to auditory stimuli, such as their ability to respond to a name call or follow simple commands. This subscale is scored on a scale of 0 to 4, with higher scores indicating better auditory responsiveness. The visual subscale assesses the ability to visually track and fixate on objects, which can provide information about visual processing abilities. This subscale is scored on a scale of 0 to 5, with higher scores indicating better visual function. The motor subscale evaluates the patient’s ability to perform movements and manipulate objects, which can provide insights into motor planning and execution abilities. This subscale is scored on a scale of 0 to 6, with higher scores indicating better motor function. The oromotor/verbal subscale assesses the ability to follow commands related to mouth movements, such as sticking out the tongue. This subscale is scored on a scale of 0 to 3. The communication subscale evaluates the ability to communicate through verbal or non-verbal means, such as gestures or eye contact. This subscale is scored on a scale of 0 to 2. The arousal subscale assesses the patient’s level of wakefulness and responsiveness to environmental stimuli, which can provide important information about overall level of consciousness. This subscale is also scored on a scale of 0 to 3. The scores from each subscale are combined to provide a total score that ranges from 0 to 23, with higher scores on the CRS-R indicating better cognitive and functional abilities. The CRS-R can be used to differentiate between various states of consciousness, such as a coma, vegetative state, minimally conscious state, or confusional state, by using the score obtained from the CRS-R assessment [5]. The CRS-R has been shown to have excellent psychometric properties, including the internal consistency, inter-rater and test–retest reliability, and diagnostic and prognostic validity for assessing the conscious state of patients with DOC [5].

### 2.3. Diffusion Tensor Imaging

Diffusion tensor imaging scanning was performed at an average of 11.44 ± 9.59 months (range 3–33.2 months) after hypoxic–ischemic brain injury. Diffusion tensor imaging data were acquired on a 1.5 T Philips Gyroscan Intera scanner (Hoffman-LaRoche, Best, Netherlands) with a six-channel head coil. For diffusion tensor imaging scans, we used a single-shot, spin-echo planar imaging method with specific imaging parameters including acquisition matrix = 96 × 96, reconstructed to matrix = 128 × 128, field of view = 221 × 221 mm^2^, repetition time = 10,726 ms, and echo time = 76 ms. In addition, a parallel imaging reduction factor of 2, an echo planar imaging factor of 49, a b-value of 1000 s/mm^2^, number of excitations = 1, and slice thickness = 2.3 mm were used. The diffusion tensor imaging data were acquired using 32 non-collinear diffusion-sensitizing gradients with 67 contiguous slices acquired in a plane parallel to the anterior commissure–posterior commissure line.

Following acquisition, the diffusion tensor imaging data analysis was performed using the Oxford Centre for Functional Magnetic Resonance Imaging of Brain (FMRIB) Software Library (FSL: www.fmrib.ox.ac.uk/fsl, accessed on 23 August 2018). To correct for the effects of head motion and image distortions, an affine multiscale two-dimensional registration method was applied. This registration method has been shown to be effective in correcting for image distortions and head motion in diffusion tensor imaging data. The brain extraction tool (BET) was performed to strip the skull from the diffusion tensor imaging data. This step was necessary to improve the accuracy of the fiber tracking by eliminating non-brain tissues that may have been included in the acquisition. A probabilistic tractography was performed using a multifiber model in FMRIB diffusion software, which allowed for fiber tracking using the BedpostX method. This method was chosen for its ability to accurately model multiple fiber orientations within each voxel, which is often observed in white matter. To analyze the microstructural properties of brain tissue, scalar maps such as fractional anisotropy and mean diffusivity were calculated using the diffusion tensor estimated from DTIFIT. These maps provide important information about the structural properties of white matter, including the directionality and integrity of neural tracts. FSLView, a software tool that allows for the visualization of scalar maps in both two-dimensional and three-dimensional views, was used to obtain visual representations of the brain. In addition, to investigate the probability of connections between regions of interests, the masks were created on the diffusion tensor imaging map using the probtracks tool. This analysis technique allowed us to examine the structural connectivity of the brain, which is essential for understanding the functional organization of the brain. The process of fiber tracking involves identifying specific regions of interest in the brain and tracing the connections between specific regions through the white matter pathways that connect them. For the tractography analysis, 5000 streamline samples were used, with step lengths of 0.5 mm and curvature thresholds set to 0.2. This comprehensive approach to diffusion tensor imaging data analysis allowed for accurate and reliable measurement of white matter fiber tracts in the brain.

### 2.4. Tractography

In each subject, the entirety and five subparts of the thalamocortical tract (the prefrontal cortex, the premotor cortex, the primary motor cortex, the primary somatosensory cortex, and the posterior parietal cortex) were reconstructed following the selection of fibers passing through various regions of interest (Figure 1) [19]. To reconstruct the thalamocortical tract, the seed and target regions of interest were placed by referring to previous studies [19,30,31,32]. The seed region of interest, which is the starting point for fiber tracking, was set on the intralaminar thalamic nuclei, while the target regions of interest were determined based on the specific anatomical regions of interest. For the prefrontal part of the thalamocortical tract, the target regions were set to include the medial prefrontal cortex as Brodmann areas 10 and 12 on the coronal image of the b0 map, dorsolateral prefrontal cortex as Brodmann areas 8, 9, and 46 on the coronal image of the b0 map, ventrolateral prefrontal cortex as Brodmann areas 44, 45 and 47 on the coronal image of the b0 map, and orbitofrontal cortex as Brodmann areas 47/12, 10, 11, and 13 on the axial image of the b0 map [19,30,31,32]. For the premotor part of the thalamocortical tract, the target region was set to the premotor cortex as Brodmann area 6 on the axial image of the b0 map [19,30,31,32]. For the primary motor part of the thalamocortical tract, the target region was placed on the primary motor cortex as Brodmann area 4 on the axial image of the b0 map [19,30,31,32]. The primary somatosensory part of the thalamocortical tract was targeted to the primary somatosensory cortex as Brodmann areas 1, 2, and 3 on the axial image of the b0 map, and for the posterior parietal part of the thalamocortical tract, the target region was set to the posterior parietal cortex as Brodmann areas 5, 7, 39, and 40 on the axial image of the b0 map [19,30,31,32].

For the analysis, the results of the thalamocortical tract, which were selected by passing fibers through the seed and target regions of interest, were visualized applying a threshold level of two streamlines through each voxel. MATLAB^TM^ (Matlab R2007b, The Mathworks, Natick, MA, USA), which is a widely used software tool for data analysis and visualization, was used to measure DTT parameters (the fractional anisotropy and the tract volume). The values of fractional anisotropy and the tract volume were determined by counting the voxels of the thalamocortical tract. This approach allowed for a more focused analysis of the thalamocortical tract, as well as the accurate quantification of important metrics such as the fractional anisotropy and the tract volume.

### 2.5. Statistical Analysis

SPSS 21.0 for Windows (SPSS, Chicago, IL, USA) was used for the statistical analysis. The correlation between the CRS-R score and DTT parameters, specifically the fractional anisotropy and the tract volume, of the entire thalamocortical tract and its five subparts, which include the prefrontal, premotor, primary motor, primary somatosensory, and posterior parietal parts, were determined by using the Spearman correlation coefficient. The significance of a detected relationship was accepted when the *p*-value of the test was <0.05. A correlation coefficient of 0.60 or greater was considered a strong correlation, a coefficient between 0.40 and 0.59 was interpreted as moderate, a coefficient between 0.20 and 0.39 was classified as weak, and a coefficient less than or equal to 0.19 was considered very weak [7].

Simple linear regression analysis with the enter method was used to further analyze the relationship between the CRS-R score and DTT parameters of the thalamocortical tract (the fractional anisotropy and the tract volume). Based on the correlation results, the DTT parameters of the thalamocortical tract that showed a significant correlation with the CRS-R score were selected as independent variables and entered into the regression model. The significance level was set at *p* < 0.05 to determine statistical significance.

## 3. Results

A summary of the correlations between the CRS-R score and the DTT parameters (the fractional anisotropy and the tract volume) for the thalamocortical tract is presented in Table 2. The CRS-R score and the tract volume of the prefrontal part of thalamocortical tract showed a moderate positive correlation (r = 0.487, *p* < 0.05). In contrast, no significant correlations were observed between the CRS-R score and the fractional anisotropy of the prefrontal part and the fractional anisotropy and the tract volume of the other four parts of the thalamocortical tract (*p* > 0.05).

The results of simple linear regression analysis for investigating the DTT parameters of the thalamocortical tract that contribute to the CRS-R score are summarized in Table 3. Among the DTT parameters in five subparts of the thalamocortical tract, the tract volume of the prefrontal part of the thalamocortical tract, which was correlated with the CRS-R score, was input as an independent variable for the dependent variable (the CRS-R). The regression model was statistically significant (*p* < 0.05), and the tract volume of the prefrontal part of thalamocortical tract accounted for 43.3% of the variance in the CRS-R score (R^2^ = 0.433). The tract volume of the prefrontal part of the thalamocortical tract was accepted in the model to explain the variability in the CRS-R score and showed a positive correlation with the CRS-R score (B = 0.015, β = 0.658, *p* < 0.05).

## 4. Discussion

In this study, DTT was used to investigate the relationship between the CRS-R score and five subparts of the thalamocortical tract in DOC patients with hypoxic–ischemic brain injury. The results are summarized as follows: (1) correlation between the CRS-R score and DTT parameters of the thalamocortical tract—a moderate positive correlation was detected between the CRS-R score and the tract volume of the prefrontal cortex part of thalamocortical tract; (2) regression between the CRS-R score and DTT parameters of the thalamocortical tract—the tract volume of the prefrontal part of thalamocortical tract could explain variability in the CRS-R score.

Regarding the DTT parameters, the fractional anisotropy is a crucial measure that provides valuable information on the organization of white matter [28,29]. The fractional anisotropy indicates the degree of directionality and the integrity of the microstructures of the white matter, making it an essential tool for detecting changes in the structural integrity of neural tracts [28,29]. Higher fractional anisotropy values are generally indicative of better white matter organization and can be used to detect alterations in the organization of neural tracts. By contrast, the tract volume refers to the total number of voxels within a neural tract, which is deemed representative of the number of neural fibers within the tract [28,29]. Changes in the tract volume reflect changes in the overall density of neural fibers, including changes in axonal density and myelination. This measure is critical in the evaluation of white matter abnormalities, as it provides information on the number and density of neural fibers present in a tract. These results denote that the CRS-R score and the tract volume of the prefrontal part of the thalamocortical tract had a more positive correlation, suggesting that the conscious state on the CRS-R is closely related to the remaining number of neural fibers of the prefrontal part in patients with hypoxic–ischemic brain injury. In addition, regression analysis showed that the tract volume of the prefrontal part of the thalamocortical tract could explain the variability of the CRS-R score. Thus, the change in the remaining number of neural fibers of the prefrontal part is related to a change in the conscious state. Hence, an increase in the number of neural fibers of the prefrontal cortex in DOC patients could facilitate the recovery of impaired consciousness in DOC patients with hypoxic–ischemic brain injury. Previous studies reported that the recovery of impairment of consciousness in DOC patients was associated with a positive change in several brain areas, including the prefrontal cortex, thalamus, hypothalamus, basal forebrain, and parietal cortex [33,34,35]. Among the above brain areas, the prefrontal cortex was the most commonly reported area. Consequently, the results of the above studies appear to coincide with these results [33,34,35].

Many studies have reported the relationship between the CRS-R score and the neural structures in DOC patients based on the results obtained by positron emission tomography, functional magnetic resonance imaging, and diffusion tensor imaging [20,21,22,23,24,25,26,27]. The brain areas related to the CRS-R score are as follows: the thalamus, lateral frontoparietal region, superior temporal region, medial frontal gyrus, primary and supplementary auditory area, occipital cortex, subcortical white matter, and a few neural tracts (thalamo-medial frontal tract, thalamo-dorsolateral prefrontal tract, and thalamo-precuneus tract) [20,21,22,23,24,25,26,27]. Regarding DTT, a few studies have reported the relationship between the CRS-R score and the thalamocortical tract in DOC patients with various brain pathologies [10,11,12,13]. In 2012, Fernández-Espejo et al. reported a positive correlation between the CRS-R score and the thalamocortical tract, which is the pathway connecting the entire thalamus with the posterior cingulate cortex/precuneus and temporoparietal junction, in 52 DOC patients with various brain pathologies (traumatic brain injury, 31 patients; hypoxic–ischemic brain injury, 10 patients; stroke, 9 patients; other, two patients) [10]. In 2017, Stafford reported that the CRS-R score had a positive correlation with the thalamocortical tract connecting the ventrolateral nucleus of the thalamus with the primary motor cortex in 15 DOC patients with a few brain pathologies (traumatic brain injury, seven patients; hypoxic–ischemic brain injury, seven patients; other, one patient) [11]. In 2019, Jang et al. demonstrated a positive correlation between the CRS-R score and the thalamocortical tract between the intralaminar thalamic nucleus and cerebral cortex in DOC 29 patients with hypoxic–ischemic brain injury [12]. During the same year, Stafford et al. (2019) reported that the thalamocortical tract connecting the ventrolateral nucleus of the thalamus with the primary motor cortex had a positive correlation with the CRS-R score in 15 DOC patients (traumatic brain injury, seven patients and hypoxic–ischemic brain injury, eight patients) [13]. However, except for one study, these studies reconstructed the thalamocortical tract differently from the present study (the thalamocortical tract connecting the thalamic intralaminar nuclei with the frontoparietal cortex). As a result, to the best of the authors’ knowledge, the present study is the first study to demonstrate a relationship between the CRS-R score and each of the five subparts of the thalamocortical tract in chronic patients with hypoxic–ischemic brain injury, using DTT.

This study had some limitations. First, the results of the present study have a generalized limitation because a small number of subjects were included. Nevertheless, in a post hoc power analysis using G*Power 3.1 (Heinrich-Heine-Universität Düsseldorf, Düsseldorf, Germany), the power (1-β error probability) was calculated as 0.92 with a sample size of 17 patients, coefficient of determination of 0.433, and α error probability of 0.05. Second, DTT analysis can overestimate or underestimate the neural fiber status in areas with crossing fibers or fiber complexity. Therefore, further studies involving a larger number of subjects and a combined analysis method with other brain mapping techniques (e.g., functional MRI and encephalography) are needed. Third, because the prefrontal cortex is a large area in the human brain, it will be necessary to identify specific areas of the prefrontal cortex. Fourth, the biomarker data for hypoxic–ischemic brain injury, such as Ubiquitin carboxy-terminal hydrolase L1 and Glial Fibrillary Acidic Protein, were not included in this study. Further studies considering the biomarkers are needed.

## 5. Conclusions

In conclusion, the prefrontal part of the thalamocortical tract was closely related to the CRS-R score in chronic patients with hypoxic–ischemic brain injury. In particular, the remaining number of neural fibers of the prefrontal part appeared to be related to the conscious state. As a result, these results suggest that the prefrontal part of thalamocortical tract could be a target for neuro-rehabilitation in DOC patients with hypoxic–ischemic brain injury. Specifically, repetitive transcranial magnetic stimulation or transcranial direct current stimulation can be applied to facilitate the prefrontal cortex, and it is expected to assist in the recovery of DOC.

## Figures and Tables

**Figure 1 healthcare-11-01148-f001:**
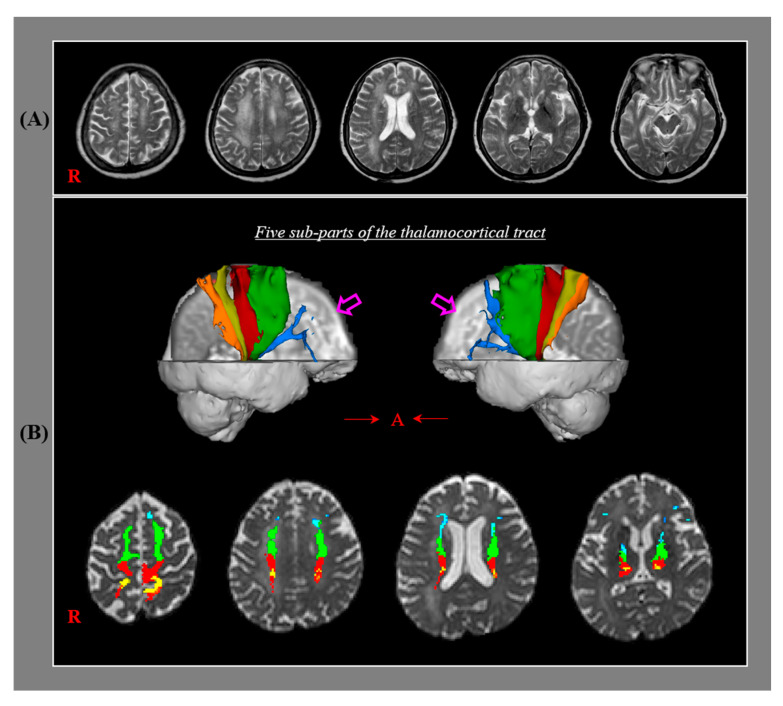
(**A**) T2-weighted brain magnetic resonance images in a representative subject (58-year-old female). (**B**) Results of diffusion tensor tractography (DTT) for the five subparts of the thalamocortical tract of the ascending reticular activating system: the prefrontal part of the thalamocortical tract (blue color), the premotor part of the thalamocortical tract (green color), the primary motor part of the thalamocortical tract (red color), the primary somatosensory part of the thalamocortical tract (yellow color), and the posterior parietal part of the thalamocortical tract (orange color). The prefrontal part is decreased compared to the other parts (arrow). R: right, A: anterior.

**Table 1 healthcare-11-01148-t001:** Demographic and clinical data of the patients with hypoxic–ischemic brain injury.

Characteristic	Patients with HI-BI (*n* = 17)
Age (years)	47.18 ± 19.01
Sex (male/female)	9/8
Duration to DTI (months)	11.44 ± 9.59
Consciousness state (coma/VS/MCS/full score)	6/3/6/2

Values represent mean ± standard deviation. HI-BI: hypoxic–ischemic brain injury, DTI: diffusion tensor imaging, VS: vegetative state, MCS: minimally conscious state.

**Table 2 healthcare-11-01148-t002:** Correlations between the score of the Coma Recovery Scale-Revised and diffusion tensor tractography parameters for the entirety and subparts of the thalamocortical tract.

		Thalamocortical Tract											
		Total		PFC		PMC		M1		S1		PPC	
		FA	TV	FA	TV	FA	TV	FA	TV	FA	TV	FA	TV
CRS-R	r	0.262	0.187	0.352	0.487	0.056	0.367	0.029	0.269	−0.024	0.197	0.310	0.329
	*p*	0.310	0.471	0.166	0.047 *	0.832	0.147	0.913	0.296	0.927	0.448	0.226	0.197

CRS-R: Coma Recovery Scale-Revised, PFC: prefrontal cortex, PMC: premotor cortex, M1: primary motor cortex, S1: primary somatosensory cortex, PPC: posterior parietal cortex, FA: fractional anisotropy, TV: tract volume. * Significant correlation between diffusion tensor parameters and clinical data (*p* < 0.05).

**Table 3 healthcare-11-01148-t003:** Simple Linear Regression Analysis for the score of the Coma Recovery Scale-Revised and diffusion tensor tractography parameters for the thalamocortical tract.

Model 1	Unstandardized Coefficients		Standardized Coefficients	*t*	*p*
	B	SE	β		
(constant)	3.743	1.668		2.244	0.040 *
TV of PFC part	0.015	0.004	0.658	3.388	0.004 *
R^2^	0.433				
*p*	0.004				

SE: standard error, TV: tract volume, PFC: prefrontal cortex. * Significant result analyzed by simple linear regression (*p* < 0.05).

## Data Availability

Not applicable.

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
