# Peer review of "Relationship between Coma Recovery Scale-Revised and the Thalamocortical Tract of Ascending Reticular Activating System in Hypoxic–Ischemic Brain Injury: A Pilot Study"

_healthcare, 2023, doi:10.3390/healthcare11081148_

Round 1
Reviewer 1 Report
Thank you for opportunity to review on the manuscript. My review opinion were as followed.
A reasonable explanation of the necessity of research is needed.
Please refer to specific research questions or research hypotheses.
In addition to the lack of research, it is necessary to complement the necessity of studying why it is necessary to find out the relationship between CRS-R and The Special Area with TCT Responsible for consciousness.
Table 1 should be revised to show the information about subjects better.
Describe the results of previous studies on the psychometric characteristics of CRS-R in method section.
There were errors. such as 2.5. subjects.
The researchers suggested the limitation which the number of subjects was small. Describe these restrictions about the possible content that can be on the results and the significance of interpreting the results of the research.
Expand the direction of future research based on the limitations of research.
It is necessary to complement the clinical or practical implications of this study.
Reviewer 2 Report
We read the article by Jang et al titled “Relationship between Coma Recovery Scale-Revised and the thalamocortical tract of ascending reticular activating system in hypoxic-ischemic brain injury”. In this work, the authors examined the relationship between the CRS-R and the five sub-parts of TCT in chronic patients with HI-BI using DTT. This was investigated in 17 patients with HI-BI patients where it was shown that the PFC part was closely related to the CRS-R score in chronic patients with HI-BI.
There is a couple of comments that the study needs to respond to:
Patients with HI-BI
1. based on limitations, where the N of patients is only 17 distributed among 3 different conscious states (vegetative, coma, and Minimally conscious state), there should be a power analysis that can justify the N used in this study and should reflect on the statistics of this study, This is not available in this work and I think this preliminary data should fall in the category of Pilot Study and the title thus should be “Relationship between Coma Recovery Scale-Revised and the thalamocortical tract of ascending reticular activating system in hypoxic-ischemic brain injury”: A Pilot Study
2. the study lacked any biomarker data for both the HI-BI such as the brain-specific such as UCH-L1 and GFAP and can guide the data analysis
3. the discussion should be more elaborate and discuss other outcomes in other neuro-related disorders such as brain injury which can strengthen the understanding and the conclusion of this work.
4. A schematic describing the five sub-parts of TCT and the state of consciousness would be helpful for the audience reading this pilot study
Round 2
Reviewer 1 Report
Thank you for your effort to revise the manuscript according to comments.
Author Response
Thank you very much.
Reviewer 2 Report
accept
Author Response
Thank you very much.